# Breaking the Shield: Analyzing and Attacking Canvas Fingerprinting Defenses in the Wild

## Abstract

Canvas fingerprinting has become one of the most effective techniques for tracking users online, allowing websites to identify and track visitors without their consent. In this paper, we investigate four primary defense techniques designed to counter canvas fingerprinting, systematically analyzing their adoption across 18 browser extensions in Chrome and Firefox, as well as built-in protections from five major browsers: Chrome, Firefox, Brave, Tor, and Safari. Our analysis reveals significant disparities in the implementation and effectiveness of these defenses, with randomization-based techniques being the most widely adopted, particularly across nine extensions and in the privacy-focused browser, Brave. Despite their sophistication, we demonstrate successful attacks on all these randomization mechanisms, revealing that their supposed non-deterministic behavior can, in fact, be predicted and exploited. In summary, we demonstrate that, unfortunately, no fully deployable defense against canvas fingerprinting attacks exists currently. We conclude by proposing recommendations to strengthen existing defenses and enhance their resistance to future attacks.

## Keywords

Security, Privacy, Online Tracking, Canvas fingerprinting Attack

## 1 Introduction

The evolution of web technologies, particularly with the introduction of HTML5 and CSS3, has revolutionized the user experience on the internet. These technologies enable developers to create dynamic, interactive, and visually appealing web pages without relying on third-party plugins. Among the significant features introduced by HTML5 is the <canvas> element, which allows developers to draw and manipulate graphics directly within the browser using JavaScript. Canvas is widely used for tasks such as rendering visualizations, interactive games, and other complex graphical operations, making it a powerful tool for web development.

Canvas fingerprinting, first studied by Mowery et al. [16], is a browser-based tracking technique that exploits subtle rendering differences in the <canvas> element to generate a high-entropy fingerprint of a user's device. This technique collects information about how a browser draws invisible or subtle content on the canvas, which can reveal characteristics specific to a user's hardware and software. While canvas fingerprinting alone may not uniquely identify a user in all cases, its ability to generate consistent, high-entropy data makes it highly effective when combined with other fingerprinting methods like fonts, plugins, and screen resolution [14]. One of the key issues with canvas fingerprinting is its passive nature, it operates without requiring any direct user interaction or consent. Trackers can invisibly collect canvas data, making it a significant privacy concern. Additionally, canvas fingerprinting poses risks beyond user privacy. As highlighted by Acharya et al. [2], attackers can leverage canvas fingerprinting to track security web crawlers, enabling them to deploy cloaking techniques that hide malicious content from these detection systems. This dual threat, targeting both users and security services, makes canvas fingerprinting a critical issue in the broader landscape of online privacy and security.

Since its inception, canvas fingerprinting has become a widely adopted tracking method across the web. A large-scale study found that approximately 14,000 of the top 1 million Alexa-ranked websites actively use canvas fingerprinting to track user [1, 5, 15]. In response to the growing concerns, both third-party developers and major browser vendors have started introducing defenses specifically targeting canvas fingerprinting. These defenses range from modifying the canvas content, either by randomizing the output or returning a fixed output, to blocking the canvas API entirely. Browsers like Firefox and Tor maintain filter lists of known third-party resources to detect and block canvas fingerprinting scripts, preventing tracking across websites. Aggressively modifying or blocking Canvas APIs has consistently led to website breakages and usability issues, as highlighted by various reports [11]. Meanwhile, filter lists struggle to keep pace with the rapidly evolving canvas fingerprinting techniques, allowing new tracking methods to slip through undetected. Given these challenges, the randomization-based technique has emerged as a more balanced and effective solution for mitigating canvas fingerprinting, offering both security and usability [12]. Brave stands out with its sophisticated 'Farbling' mechanism [10], which leverages a randomization-based technique to balance security and usability in defending against canvas fingerprinting. Despite these efforts, the robustness of such defenses has not been thoroughly tested against potential attacks.

In this paper, we address this gap by proposing the first set of attacks specifically targeting randomization-based defenses. We focus on 9 popular extensions that implement randomized output techniques and the "Farbling" mode in Brave browser. Our contributions are as follows:

**1.Canvas fingerprinting Defenses.** We conduct a comprehensive investigation of current defense techniques against canvas fingerprinting proposed by researchers, browser vendors, and privacy advocates.

**2.Canvas fingerprinting Defense Adoption.** We systematically analyze 18 extensions from the Chrome and Firefox web stores, as well as the built-in protections of 5 major browsers (Chrome, Firefox, Brave, Tor, and Safari), to understand how these techniques are being adopted in the wild.

**3.The Pixel-Recovery and Statistical Attack.** We successfully attack the most effective defenses, the randomization-based techniques, demonstrating vulnerabilities in nine extensions and the Brave browser's 'Farbling' mechanism to our two proposed attack vectors: the Pixel-Recovery Attack and Statistical Attack.

**4.Recommendations.** We offer recommendations to strengthen existing defenses and provide insights into how these techniques can be made more resilient against sophisticated attacks.

Finally, we plan to release the code implementations for our two proposed attacks to raise awareness of this issue and encourage further research in online tracking defenses

## 2 Background

The introduction of the `<canvas>` element was a major development in web technology, enabling browsers to perform a range of graphical tasks natively, without external plugins. Introduced by Apple in 2004 and standardized in HTML5, the canvas provides a programmable drawing surface for rendering dynamic visual content [23]. Using JavaScript, developers can create 2D graphics, animations, interactive visualizations, and even 3D scenes with WebGL [8]. The canvas operates as a bitmap, allowing developers to manipulate individual pixels. APIs like `getContext()` provide drawing capabilities, while methods like `getImageData()` extract pixel data in arrays. Each pixel has four RGBA values (0–255), so a 500x400 canvas generates 800,000 data points (500 `width` × 400 `height` × 4 channels). These extracted values are what enable canvas fingerprinting. Minor differences in factors like font rendering, anti-aliasing, and hardware acceleration produce a unique digital fingerprint. While it might not be sufficient on its own to track users, it significantly contributes to the uniqueness of a user's fingerprint when combined with other techniques such as User-Agent or Fonts. Among browser fingerprinting methods, canvas fingerprinting generates some of the highest entropy, making it a powerful tool for tracking users across websites [14, 15].

Studies have consistently highlighted its widespread use and effectiveness. Mowery and Shacham's "Pixel Perfect" demonstrated how canvas rendering variations could generate distinct fingerprints for each device [16]. Subsequent large-scale studies by Acar et al. and Engelhardt et al. revealed that canvas fingerprinting was actively exploited by thousands of websites in the Alexa top 100,000 and top 1 million, respectively, showing its persistence and growing adoption as a tracking mechanism [1, 5]. These findings emphasize canvas fingerprinting's resilience and its attractiveness to online trackers.

The increasing prevalence of canvas fingerprinting has raised significant privacy concerns. The ability to uniquely identify and track users when combined with other techniquess without their consent poses a threat to user privacy. In response, various browser vendors and privacy-focused communities have developed mitigation strategies. For example, the Brave browser has introduced privacy updates that actively pIn additionrevent canvas fingerprinting by blocking scripts and providing randomized or fixed canvas outputs [4]. Similarly, Mozilla's Firefox browser has implemented fingerprinting protection mechanisms, including prompts for user consent before allowing the canvas API to be accessed [18]. These efforts represent a growing recognition of the need to protect users from the invasive tracking capabilities enabled by canvas fingerprinting.

In addition, security companies frequently rely on web crawlers to identify and analyze phishing websites and other social engineering attack platforms. These web crawlers serve as automated tools to detect malicious content and ensure the safety of users. However, attackers have adapted their strategies by employing browser fingerprinting techniques, including canvas fingerprinting, to differentiate between human visitors and security crawlers. This enables attackers to perform cloaking attacks, where they present benign content to crawlers while delivering malicious content to actual users [2]. As a result, canvas fingerprinting not only poses privacy risks to regular users but also undermines the effectiveness of security crawlers by allowing malicious websites to evade detection. This adds an additional layer of complexity to web security, necessitating more sophisticated countermeasures to detect cloaking attempts and defend against such evasion tactics.

## 3 Defense Strategies

Canvas fingerprinting poses a significant privacy threat due to its wide adoption and effectiveness. Several techniques have been developed and deployed to mitigate the risks associated with this tracking method. In this section, we categorize common defense techniques used by researchers, browser vendors, and privacy advocates to protect users against canvas fingerprinting.

### 3.1 Blocking APIs and Access

Blocking access to the Canvas API is a direct method for defending against canvas fingerprinting. By entirely disabling the Canvas API, browsers prevent websites from collecting device-specific rendering data, effectively eliminating the possibility of canvas-based fingerprinting. This approach aims to enhance user privacy by restricting the data available to trackers.

While effective, blocking the Canvas API presents significant usability challenges. The API is essential for rendering dynamic graphics, animations, and real-time visualizations, which are integral to many websites, including gaming platforms, data visualization tools, and interactive applications. Consequently, disabling the API can lead to broken features and a degraded user experience, as critical elements may fail to load or function correctly. For example, banking websites that rely on the Canvas API for authentication mechanisms may encounter disruptions, compromising both security and usability [1].

This creates a trade-off between privacy and usability: privacy-conscious users, such as those using Tor Browser, may accept the impact on website functionality as a necessary compromise. However, for mainstream users, the disruption caused by blocking canvas functionality can outweigh the privacy benefits, especially given the increasing reliance on Canvas in modern web applications. This tension highlights the need for privacy defenses that protect user data while ensuring seamless usability across web platforms.

### 3.2 Modifying the Canvas Content

Modifying canvas pixel data is a widely used technique to counter canvas fingerprinting while preserving the functionality of the Canvas API. This defense is typically implemented through two primary methods: randomizing the canvas output or returning a fixed canvas. Both techniques aim to prevent trackers from generating consistent fingerprints, but they do so in fundamentally different ways.

**Randomized Output.** The first method introduces subtle, random variations in the canvas pixel data every time a script accesses it. This process perturbs pixel values specifically, the RGBA channels while ensuring that the visual content remains unchanged

for the user as shown in Figure 2 in the Appendix. The goal is to make the canvas output undeterministic and irreversible, meaning each fingerprint is unique across sessions, thus preventing trackers from correlating user activity. However, this technique requires thorough implementation to be effective. It requires careful tuning to avoid noticeable distortions in visual content; too much randomization could degrade legitimate rendering, while insufficient variation may still allow trackers to identify patterns. Additionally, randomization can sometimes inadvertently signal to trackers that privacy protections are in place, potentially making the user more conspicuous. Therefore, to maintain both properties, undeterministic and irreversible, it is crucial that randomized output is implemented correctly. If not executed properly, its effectiveness may be compromised.

**Fixed Output.** The second method involves returning a consistent, blank, or fixed canvas output. By fixing the canvas output, all users appear identical to the tracker, effectively eliminating the uniqueness that fingerprinting relies upon. Unlike randomization, where output varies each time, fixed output provides a static fingerprint that is shared by all users. This method ensures that no session-specific data can be extracted. While it is a simpler and more consistent defense, it can disrupt the functionality of websites that rely on canvas rendering, potentially leading to broken elements or reduced interactivity.

### 3.3 Filter Lists

Filter-list-based blocking mechanisms provide an additional layer of defense against canvas fingerprinting by targeting known third-party tracking domains and preventing their fingerprinting scripts from executing when a webpage is loaded. These filters operate by identifying and blocking requests to domains associated with known trackers, effectively stopping many widely-documented tracking techniques from being deployed [6].

Despite their effectiveness against well-established and frequently updated scripts, filter lists have inherent limitations. They struggle to detect and block newly developed or obfuscated fingerprinting scripts that haven't yet been added to the lists. These methods are inherently reactive, requiring constant updates to remain effective. As a result, advanced or rapidly evolving fingerprinting techniques can bypass these defenses until the filter lists are updated to recognize the new threats. Nonetheless, filter-list-based defenses remain widely used due to their ease of use, minimal impact on web performance, and their ability to mitigate many common tracking methods with little disruption to the user experience.

### 3.4 Machine Learning Models

Machine learning (ML) models provide adaptive defenses against canvas fingerprinting by detecting both known and new tracking methods. Unlike static filters or randomization, ML models are trained to recognize patterns of fingerprinting attempts. Studies like FP-Radar by Bahrami et al., and work by Iqbal et al. and Reitinger and Mazurek, highlight ML's effectiveness in real-time detection [3, 13, 21]. These models analyze large web traffic datasets, identifying even unknown scripts. ML's adaptability is a key strength, but it requires significant computational resources, may slow page loading, and risks false positives, limiting widespread adoption.

## 4 Defense Strategies Adoption

In this section, we investigate the adoption of canvas fingerprinting defense techniques by examining both third-party browser extensions and the native protections offered by popular browsers. We analyze the functionality and effectiveness of these defenses, providing a comprehensive view of the current landscape and how these methods are being used to combat canvas fingerprinting in practice.

### 4.1 Extensions for Canvas Fingerprinting Defense

We began by conducting an extensive search for canvas fingerprinting defense extensions in the Firefox and Chrome web stores. Using keyword searches such as "canvas fingerprinting protections" and "canvas fingerprinting defenses," we identified 18 extensions designed to mitigate canvas fingerprinting attacks. These extensions are listed and summarized in Table 1. For each extension, we analyzed its description and code to understand the defense techniques employed. Of the 18 extensions, 12 implemented techniques that modify the canvas content, with 9 using randomized output. The popularity of this method suggests its effectiveness in mitigating fingerprinting. Only 3 extensions block the Canvas API altogether. Extensions like *uBlock Origin*, *Adblock Plus*, and *Privacy Badger* use Filter Lists to block known third-party tracking sites, adding an extra layer of defense against canvas fingerprinting.

### 4.2 Built-in Protections in Major Browsers

In addition to third-party extensions, we evaluated the built-in protections offered by five major browsers: Safari, Chrome, Firefox, Brave, and Tor. We thoroughly reviewed the official documentation for each browser to confirm the presence and efficacy of any canvas fingerprinting defenses. Table 2 summarizes our findings.

**Safari.** Safari employs randomization as a core defense for tracking, primarily for fonts and plugins. However, it lacks built-in protections specifically against canvas fingerprinting [9].

**Chrome.** Chrome, like Safari, does not offer significant built-in protections against canvas fingerprinting. Instead, users must rely heavily on third-party extensions to block tracking attempts.

**Firefox.** Firefox provides robust anti-fingerprinting protections through fixed output, blocking, and filter list mechanisms. A notable feature is the *randomDataOnCanvasExtract* setting, which returns either randomized data or a blank canvas, thwarting fingerprinting attempts by making extracted data unusable. Additionally, Firefox's **Enhanced Tracking Protection (ETP)** allows users to block access to the Canvas API entirely, preventing websites from reading canvas data for tracking [7]. Firefox also employs regularly updated filter lists to block known fingerprinting scripts, preemptively stopping many common techniques.

**Tor.** Built on Firefox ESR (Extended Support Release), Tor Browser offers enhanced privacy protections, including canvas fingerprinting defenses like blocking, fixed output, and filter lists. Its primary defense is fully blocking the Canvas API, preventing websites from accessing any canvas data and rendering fingerprinting attempts ineffective. This strong focus on anonymity comes with reduced functionality on some websites, reflecting Tor's priority of privacy

over usability [19]. Given its role as a tool for anonymity, even minor data leaks are treated as significant risks.

**Brave.** Brave Browser is known for its advanced privacy protections, particularly in the realm of fingerprinting defenses. One of its key innovations is the "Farbling" technique, which is Brave's implementation of randomized output technique to prevent canvas fingerprinting. "Farbling" Default mode introduces slight variations in canvas data during fingerprinting attempts, ensuring that websites are unable to generate consistent identifiers for tracking purposes. In addition to randomizing the canvas content, Brave has also employed a fixed output mechanism in "Farbling" Maximum mode.

Besides "Farbling" mechanism, Brave offers a strict blocking mode that prevents any JavaScript code from accessing canvas elements, effectively blocking the Canvas API. While this strict blocking provided strong protection, it caused significant usability issues on many websites that rely on canvas rendering for interactive features. As a result, Brave has announced plans to retire its strict blocking mode in favor of less intrusive options that balance usability with privacy. This move reflects Brave's recognition that blocking the Canvas API entirely may not be a sustainable solution for mainstream users who need to access rich, interactive websites [11, 12]. Instead, Brave has shifted towards "Farbling" as a way to offer strong fingerprinting defenses without severely disrupting web functionality.

## 4.3 Adoption Trends

Our analysis highlights the techniques employed in the defense against canvas fingerprinting, each with its own strengths and limitations. Blocking the Canvas API, while offering robust protection, is less commonly adopted due to its disruptive impact on website functionality. Implemented by three extensions and browsers like Firefox, Brave, and Tor, this method can break interactive features on many sites, leading to a frustrating user experience. Brave's decision to phase out strict blocking exemplifies the usability trade-offs inherent in this approach [12].

Fixed output, utilized by three extensions and browsers, offers a more rigid defense by standardizing canvas output across users. Although this technique can effectively mitigate some fingerprinting attempts, it often leads to compatibility issues on sites that rely heavily on canvas elements for interactive content, resulting in a loss of functionality that can deter users.

Filter lists, popular in extensions such as *uBlock Origin*, *Adblock Plus*, and *Privacy Badger*, provide a lightweight, user-friendly alternative by blocking known fingerprinting scripts. However, their dependence on predefined lists limits their ability to detect new or obfuscated techniques, making them less effective against the constantly evolving landscape of fingerprinting threats.

Amidst these various approaches, the randomized output technique emerges as the most promising and widely adopted solution, employed by nine extensions and the privacy-focused broswer Brave. This technique excels in providing robust protection against fingerprinting without significantly impacting usability. By introducing subtle variations in canvas data, randomized output effectively disrupts consistent tracking while allowing websites to function normally. Brave's "Farbling" mechanism, which sophisticatedly

randomizing the canvas output, stands out as a particularly effective implementation of this technique. Brave plans to continue refining this approach, aiming to enhance privacy protections while maintaining a seamless user experience [12]. As the landscape of canvas fingerprinting evolves, the randomized output technique positions itself as a key player in the fight against tracking, successfully balancing strong defenses with usability.

## 5 Experiments

This section describes the experiments conducted to evaluate the effectiveness of both browser extensions and built-in browser features in mitigating canvas fingerprinting. The primary goal was to determine whether these defenses performed as advertised. The results, indicating which protections worked as expected and which failed, are summarized in Tables 1 and 2.

## 5.1 Experimental Setup

To test these protections, we created a test page with two types of canvas elements, as illustrated in Figure 1:

A **Filled canvas**, featuring graphical content inspired by FingerprintJS [22], a commonly used fingerprinting library.

A **Base canvas**, with a simple background color of RGBA(150, 150, 150, 0.5), designed to evaluate how defenses handle canvas elements without complex content.

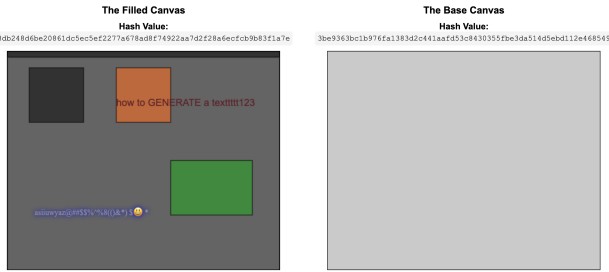

**Figure 1: Example of Filled and Base Canvas**

We extracted the content from each canvas using the *getImageData()* API, which returns an *ImageData* object containing pixel data in an array format [17]. To verify the integrity of the protection mechanisms, we computed a hash for the canvas content and monitored any changes when protections were activated or deactivated. Each protection was evaluated based on its core technique Randomized Output, Fixed Output, Blocking, and Filter Lists. We compared the canvas content before and after enabling these protections by observing the corresponding hash values.

## 5.2 Evaluating Canvas Protection Techniques

**Randomized Output:** For extensions and browsers using randomized output techniques, we examined the base canvas. Without protection, the Base canvas consistently returned pixel values of 150 for all RGB components. Once randomization was enabled, we observed that the pixel values and hashes varied across page reloads, confirming that randomization disrupted consistent fingerprint generation.

| Extension | Browser | | Modifying | | Blocking | Filter List | Status |
|---|---|---|---|---|---|---|---|
| | Chrome | Firefox | RO | FO | | | |
| Canvas Blocker - Fingerprint Protect | ✓ | ✓ | | ✓ | | | ✗ |
| Canvas Finger Defender by Ilgur/yubi | ✓ | ✓ | ✓ | | | | ✓ |
| CanvasFingerprintBlock | ✓ | | | ✓ | | | ✓ |
| Fingerprint Spoofing | ✓ | | ✓ | | | | ✓ |
| Trace | ✓ | ✓ | ✓ | | | | ✓ |
| CyDec Security Anti-FP | ✓ | ✓ | ✓ | | ✓ | | ✗ |
| Browque | ✓ | | ✓ | | | | ✓ |
| Canvas Blocker for Google Chrome | ✓ | | ✓ | | | | ✗ |
| Don't Fingerprint Me | ✓ | | | | | ✓ | ✗ |
| Privacy Extension | ✓ | | | | ✓ | | ✗ |
| Canvas Defender by Multilogin | ✓ | | ✓ | | | | ✓ |
| Fingerprint Shield by Francesco De Stefano | | ✓ | | ✓ | | | ✓ |
| Canvas Blocker by kkapsner | | ✓ | | | ✓ | | ✓ |
| PeyTy's Browser Privacy by PeyTy | | ✓ | ✓ | | | | ✓ |
| WebGL Fingerprint Defender | ✓ | | ✓ | | | | ✓ |
| uBlock Origin | ✓ | ✓ | | | | ✓ | ✓ |
| Adblock Plus | ✓ | ✓ | | | | ✓ | ✓ |
| Privacy Badger | ✓ | ✓ | | | | ✓ | ✓ |

**Table 1: Details of Canvas Defender Extensions and Their Supported Techniques. Note: RO = Randomized Output, FO = Fixed Output. Status indicates whether the extension works as expected.**

| Technique | Safari | Chrome | Firefox | Brave | Tor |
|---|---|---|---|---|---|
| Blocking | | | ✓ | ✓ | ✓ |
| RO | | | | ✓ | |
| FO | | | ✓ | ✓ | ✓ |
| Filter List | | | ✓ | | ✓ |
| Status | 🔴 | 🔴 | ✓ | ✓ | ✓ |

**Table 2: Canvas Fingerprinting Defense Techniques in Major Browsers. Note: RO = Randomized Output, FO = Fixed Output, Status indicates whether the browser's fingerprinting defenses work as expected.**

**Fixed Output:** When testing fixed output protections, we first captured the base canvas content without defenses and then re-tested with protections enabled. As expected, the pixel values were replaced by random values, such as 0 across the entire canvas. We tested across different devices and browsers, and since all pixel values were set to 0, the hash remained consistent, indicating that the protection enforced a stable, unchanging canvas output.

**Blocking:** For protections that blocked access to the Canvas API, we used the *getImageData()*, *toDataURL()*, and *toBlob()* APIs to attempt canvas data extraction. When blocking was enabled, these API calls returned either errors or empty results, effectively preventing the retrieval of canvas content. We were unable to produce any hashes for both types of canvases.

**Filter Lists:** Filter-list-based protections were tested by visiting websites known to use third-party tracking scripts [22]. Additionally, we embedded FingerprintJS within the filled canvas to determine if these protections could identify and block fingerprinting attempts.

## 5.3 Results

The experimental results are detailed in Tables 1 and 2. Out of the 18 extensions tested, 5 were found to be ineffective, failing to provide the claimed protection against canvas fingerprinting. These extensions neither blocked nor modified canvas content as expected, revealing gaps in their functionality and reliability. Conversely, 13 extensions performed as advertised, successfully modifying or blocking canvas content and providing consistent protection. These extensions demonstrated robust, reliable functionality, aligning with their intended privacy goals.

Regarding built-in browser protections, major browsers such as Firefox, Brave, and Tor worked as expected. These browsers consistently applied the intended protections, such as randomization or blocking, without any significant issues. We also tested whether Fixed Output and Blocking techniques affected legitimate website functionality by visiting various canvas-based sites, identified via source code search engines [20], including YouTube and web applications for painting or editing. In some cases, these sites malfunctioned with protections enabled, though the issues were inconsistent, suggesting that while the impact on usability is unpredictable, it is not negligible

## 6 Pixel-Recovery Attack on Browser Extensions

Blocking, Fixed Output, and Filter Lists defenses all have significant limitations in canvas fingerprinting, often compromising website functionality or failing to detect advanced fingerprinting scripts. In contrast, Randomized Output has emerged as the preferred technique, implemented by nine out of 18 analyzed extensions and Brave. However, despite its growing popularity, comprehensive testing of Randomized Output against targeted attacks remains limited. In this section, we propose the **Pixel-Recovery** attack aimed at assessing the effectiveness of Randomized Output techniques implemented by the browser extensions.

Through an analysis of their published source code and experiments performed as described in Section 5, we observed that none of these implementations fully realized the intended properties of the Randomized Output technique. Ideally, this method should produce unpredictable, non-deterministic outputs that cannot be reversed. In practice, all nine extensions exhibited patterns that allowed us to reconstruct the original canvas content.

All nine extensions follow a similar, basic implementation to randomize the output of canvas elements. At the start of each session, the extension generates a random tuple of RGBA values, denoted as $P(R, G, B, A)$, representing a perturbation vector for a pixel. This perturbation is applied to specific sections of the canvas (e.g., a corner or predefined area), or in some cases, to the entire canvas. The perturbation process typically involves basic arithmetic operations such as addition, subtraction, or XOR. For example, if a pixel originally has the value $A(137, 196, 245, 0.5)$ and the perturbation vector is $P(3, 4, 5, 0.1)$, the pixel value is transformed into $A'(140, 200, 250, 0.6)$ as illustrated in Figure 2 in the Appendix. This transformation is applied uniformly across all selected pixels, with each extension varying the region where the perturbation is applied. While these extensions differ in how they select regions of the canvas for perturbation, the underlying randomization process is consistent enough across all nine that a generalized attack can be formulated. We demonstrate that it is possible to recover the original canvas output by determining the perturbation values $P(R, G, B, A)$ used in the transformation, and then reversing their effects.

We designed a test page to exploit this flaw by generating two canvases: a **Filled Canvas** and a **Base Canvas**. As shown in Figure 1, the Base Canvas is initialized with uniform pixel values of $RGBA(150, 150, 150, 0.5)$ before the page is loaded. This known, uniform base provides a reliable benchmark for identifying the perturbation vector. The Filled Canvas is designed to simulate the kind

**Data:** Generate one Base Canvas $B$ and one Filled Canvas $F$
**Result:** Reconstruct original content of the $F$ Canvas
1 - Extract pixel data $F_{\text{pixels}}$ from $F$ using `getImageData()`;
2 - Compute the hash value $H(F)$ for $F_{\text{pixels}}$;
3 - Activate the extension to modify both $B$ and $F$;
4 - Extract pixel data $B_{\text{pixels}}$ and $F_{\text{pixels}}$ from both $B$ and $F$ after modification;
**foreach** *pixel p in B$_{pixels}$* **do**
  Compute perturbation as:
    $P(R, G, B, A) = p_{\text{modified}} - RGBA(150, 150, 150, 0.5)$;
**end**
**foreach** *pixel q in F$_{pixels}$* **do**
  Reverse the perturbation:
    $q_{\text{original}} = q_{\text{modified}} - P(R, G, B, A)$;
**end**
5 - Compute the hash $H(F')$ for the reconstructed canvas $F'_{\text{pixels}}$;
**if** $H(F') = H(F)$ **then**
  Reconstruction of the original canvas content is successful;
**end**

**Algorithm 1:** The Pixel-Recovery Attack PseudoCode

of content typically used for tracking purposes, containing more complex images and a variety of pixel values that are often targeted by fingerprinting scripts. This allows us to assess the effectiveness of our attack.

Given that the values in the Base Canvas were predefined and identical across all pixels, we systematically iterated through the extracted pixel data to determine the perturbation vector $P(R, G, B, A)$ applied by the extension. This process involved comparing the extracted pixel data with the predefined $RGBA(150, 150, 150, 0.5)$ and calculating the perturbed differences applied by the extension. Once the perturbation vector was identified, we applied the reverse operation to the Filled Canvas, effectively undoing the randomization.

By reversing the perturbations on the Filled Canvas, we were able to recover the original pixel values and compute a hash that matched the unperturbed canvas. To further validate the consistency of our attack, we reloaded the page ten times with the extensions enabled. Each time, the hash for the Filled Canvas remained consistent, demonstrating that our method reliably bypassed the randomization. The hash of the Base Canvas, however, varied with each reload, as expected, due to the randomization process reapplying different perturbations on each load. We also describe details our Pixel-Recovery Attack in the Algorithm 1

This attack reveals that, while the Randomized Output technique is widely adopted, its current implementations across these nine extensions fail to achieve the desired non-determinism. By analyzing the perturbation process and reversing it, we demonstrate that the original canvas content can be reliably recovered, even in the presence of randomization.

# 7 Statistical Attack on Farbling Mechanism

Brave has developed a defense mechanism known as **Farbling** to combat browser fingerprinting including canvas fingerprinting, particularly focusing on the randomization of semi-identifying browser features such as canvas fingerprints. The goal of Farbling is to make it challenging for websites to track users based on their browser fingerprints, while maintaining usability for benign websites. Farbling introduces controlled randomness into these browser features, ensuring that fingerprinting attempts cannot produce consistent results across sessions or websites. Brave's Farbling has three modes:

**1.Off:** No fingerprinting protections are applied.

**2.Default:** This mode implements a *Randomized Output* technique for canvas fingerprinting, designed to minimize the risk of breaking websites.

**3.Maximum:** In this mode, Brave uses a *Fixed Output* technique that returns the same canvas data for all fingerprinting attempts. However, due to usability issues and site breakage caused by this technique, Brave has deprecated this mode in favor of more user-friendly approaches [11, 12].

For this analysis, we focus on Brave's Default mode, where the **Farbling Mechanism** operates through Randomized Output, which strikes a balance between security and usability.

## 7.1 Farbling Mechanism in Brave

Brave's open-source nature allows for detailed insights into its canvas fingerprinting defense mechanism, known as Farbling . Farbling introduces randomization into canvas data to prevent consistent fingerprinting across browsing sessions, domains, and even different canvas content within the same session.

When a canvas extraction API like `getImageData()` is invoked, Brave generates a unique 32-byte "canvas key". This key is calculated based on three factors: the session key, the domain name, and the content of the canvas itself. The session key ensures that the randomization is consistent within the same session but differs between sessions, while the canvas content introduces further variability, even when other factors like the session and domain remain unchanged. Brave only perturbs one of the color channels—Red, Green, or Blue—determined by processing the first byte of the domain name and taking the result modulo 3. The Alpha channel (opacity) is left unchanged. For instance, if the result of the modulo operation is 0, the Red channel is perturbed. The actual perturbation is performed using the canvas key. Brave processes each byte of the key, starting with the least significant bit (LSB). For each bit, it selects a pixel and applies an XOR operation between the bit and the pixel's preselected color channel. If the bit is 1, the color channel is changed slightly, while if it is 0, the channel remains the same.

To ensure the perturbations are spread across the canvas in a seemingly random but reproducible way, Brave uses a Linear Feedback Shift Register (LFSR) to select the next pixel to perturb. The LFSR ensures that the pixels are chosen in a pseudo-random sequence, preventing any easy prediction of which pixels will be changed. This process repeats across all 32 bytes of the key, affecting up to 256 pixels. The perturbations are scattered across the canvas in a pseudo-random manner, making it difficult to extract

**Data:** $session\_key, domain\_key, canvas\_content$
**Result:** Perturbed $canvas\_content$
1 - Extract the first byte of the domain key: $first\_byte$;
2 - Select the perturbation channel: $channel = first\_byte\%3$;
3 - Compute
 $session\_plus\_domain\_key = session\_key \oplus domain\_key$;
4 - Compute 32-byte $canvas\_key = HMAC-$
 $SHA256(session\_plus\_domain\_key, canvas\_content)$;
5 - Initialize seed $v = $ (uint64_t) $canvas\_key[0:8]$;
6 - Initialize $pixel\_count = canvas\_width * canvas\_height$;
**foreach** $byte\ key\_byte\ in\ canvas\_key$ **do**
  **for** $bit\ position\ j = 7\ to\ 0$ **do**
    Extract LSB from $key\_byte$: $lsb = key\_byte \& 0x1$;
    Compute
     $pixel\_index = 4 \times (v\%pixel\_count) + channel$;
    Perturb:
     $pixels[pixel\_index] = pixels[pixel\_index] \oplus lsb$;
    Shift $key\_byte$ right: $key\_byte = key\_byte >> 1$;
    Update seed $v$: $v = lfsr\_next(v)$;
  **end**
**end**

**Algorithm 2:** Brave's Farbling Mechanism

consistent fingerprints from the canvas across sessions or domains. For detailed technical steps, refer to Algorithm 2.

**Challenge of Attacking Brave's Farbling.** Our Pixel-Recovery attack on the extensions in Section 6 proves ineffective against Brave's Farbling mechanism. This is because Brave's per-session canvas key ensures that even within the same domain and session, each canvas is perturbed differently due to variations in the canvas content itself. As a result, it is not possible to reverse the perturbations applied to the Filled Canvas by using the Pixel-Recovery attack.

## 7.2 Statistical Attack Methodology

This section introduces the **Statistical Attack**, aimed at evaluating the resilience of Brave's Farbling mechanism. While Farbling introduces random perturbations to canvas data to hinder consistent fingerprinting, this randomness proves insufficient for thwarting fingerprint reconstruction. Our attack exploits this limitation by carefully designing an experiment involving the generation of **five** Filled Canvases, each with dimensions of 500x400 pixels. To introduce controlled variation across these canvases, we add a 500x10 rectangle with a different color at the top of each canvas as shown in Figure 3 in the Appendix, ensuring that the perturbation mechanism in Brave's Default mode is triggered. Despite these controlled variations, Brave perturbs each canvas differently as expected. We focus on the part of the canvas excluding the top rectangle and extract the content using the `getImageData()` API, resulting in arrays of **800,000 numbers** for each canvas (calculated as 500 pixels (width)×400 pixels (height)×4 color channels per pixel).

To reconstruct the original canvas content, we applied a **majority voting** technique. For each pixel channel across the five arrays, we observe which values appear most frequently. Since Brave's Farbling mechanism perturbs only one preselected channel

**Data:** Generate 5 Filled Canvases $F_1, F_2, F_3, F_4, F_5$
**Result:** Reconstruct original content of the $F_1$ Canvas
1 - Add a 500x10 unique-color rectangle to the top of each
  canvas;
2 - Extract pixel data $F_{1_{\text{pixels}}}$ from the main canvas $F_1$
  (excluding the top rectangle) using `getImageData()`;
3 - Compute the hash value $H(F_1)$ for the $F_{1_{\text{pixels}}}$;
4 - Activate Farbling to perturb all 5 Filled Canvases;
**foreach** *pixel p in the pixel arrays* **do**
    **foreach** *channel c in R, G, B (channels)* **do**
        Count occurrences of *value* across the 5 canvases;
        **if** *value* $\geq 3$ **then**
            Select as original value for channel $c$ of pixel $p$;
        **end**
    **end**
**end**
5 - Compute the hash values $H(F_1')$ for the reconstructed
  canvas $F_{1_{\text{pixels}}}'$;
**if** $H(F_1') = H(F_1)$ **then**
    Reconstruction of the original content is successful;
**end**

**Algorithm 3:** The Statistical Attack PseudoCode

(R, G, or B) for each pixel, the other channels remain consistent. By counting the number of times each value appears across the five canvases, we deduce the most common value and infer that this was the original, unperturbed value. Specifically, if a number appears three or more times in the five arrays, we confidently select that value as the original as demonstrated in Figure 4 in the Appendix. For context, each canvas contains 200,000 pixels, and each pixel has 4 channels (R, G, B, A), resulting in an array of 800,000 numbers per canvas. Across five canvases, we work with a total of $5 \times 800,000 = 4,000,000$ numbers. The key assumption here is that the probability of any pixel channel being perturbed more than half the time (i.e., appearing in 3 or more canvases) is exceptionally low, and our majority voting technique allows us to recover the original values effectively. We detail our Statistical Attack in Algorithm 3.

We also calculate the probability of a specific channel being perturbed more than half the time across five canvases to demonstrate the certainty in our attack. The likelihood of a pixel channel being selected for perturbation 3 or more times out of the five canvases is given by the binomial distribution: To determine the probability of selecting the same pixel channel three or more times across five canvases, we use the binomial distribution. Given the total number of canvases $n = 5$ and the probability of selecting a specific pixel channel $p = \frac{1}{800,000}$:

$$P(X \geq 3) = 1 - \sum_{k=0}^{2} P(X = k)$$

Where:

$$P(X = k) = \binom{n}{k} p^k (1-p)^{n-k}$$

Thus, the probability of selecting the same channel 3 or more times is:

$$P(X \geq 3) = 1 - P(X < 3) \approx 1.953 \times 10^{-17}$$

This probability is extremely close to zero, confirming our intuition that selecting the same pixel channel across five canvases is unlikely. We also calculate the minimum number of canvases required to reliably reconstruct the original canvas content without failure. Using similar binomial probability calculations, we determine that generating **4 canvases** is sufficient to apply majority voting and recover the original values. However, generating **5 canvases** provides a stronger guarantee of success, ensuring that even in worst-case scenarios, our attack remains effective with a probability of failure that approaches 0, as demonstrated above.

## 8 Disclosure and Recommendation

**Recommendation.** Based on our analysis of the attacks on Brave's Farbling mechanism and the randomization techniques used by 9 extensions, we offer two recommendations to enhance the effectiveness of canvas fingerprinting defenses. First and foremost, any implementation of randomization should ensure two essential properties: undeterministic and irreversible. Undeterministic randomization means that the output should vary with each execution, thereby preventing trackers from establishing consistent patterns. Irreversible randomization entails that the original data cannot be reconstructed from the modified output. Additionally, it is crucial for browser vendors and extension developers to implement monitoring mechanisms for API calls. By tracking the usage patterns and frequency of canvas API calls, developers can identify suspicious behavior indicative of tracking attempts. This could involve setting thresholds for the number of times a canvas element is accessed within a certain timeframe, allowing for the detection of potential fingerprinting attempts.

**Disclosure.** We disclosed our findings to Brave, which acknowledged that our attack reintroduces some entropy into the browser's fingerprinting capabilities. In response, Brave plans to limit the number of Canvas API read-backs between user activations, aligning with our recommendations to mitigate future misuse. Furthermore, Brave recognizes the necessity of staying abreast of emerging fingerprinting attacks to constantly improve their defenses and implement additional protections for users

## 9 Conclusion

In this paper, we provide a thorough investigation into canvas fingerprinting defenses, focusing on 18 extensions from the Chrome and Firefox web stores and the built-in protections of five major browsers: Chrome, Firefox, Brave, Tor, and Safari. Our analysis reveals vulnerabilities in widely adopted randomization techniques, as we successfully attacked nine extensions and the Brave browser's "Farbling" mechanism. These findings underscore the need for ongoing enhancements in privacy technologies to counteract evolving tracking methods. Additionally, we offer targeted recommendations to strengthen existing defenses, highlighting the importance of maintaining robust, user-friendly solutions. As canvas fingerprinting continues to rise, our work emphasizes the critical need for effective privacy measures that empower users to navigate online without intrusive tracking.

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

# A Examples

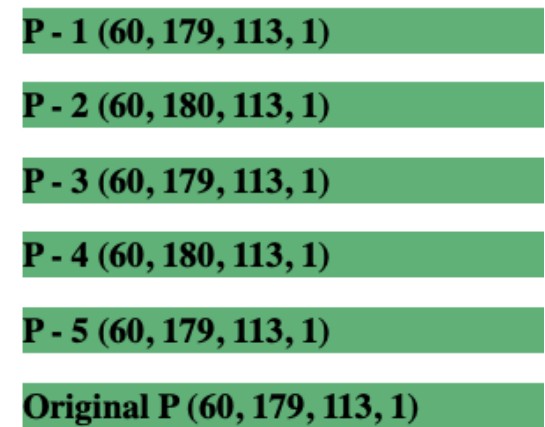

Figure 2: An example of the pixel perturbation process illustrates how the underlying values of a pixel change after perturbation is applied, yet these alterations remain visually imperceptible to users.

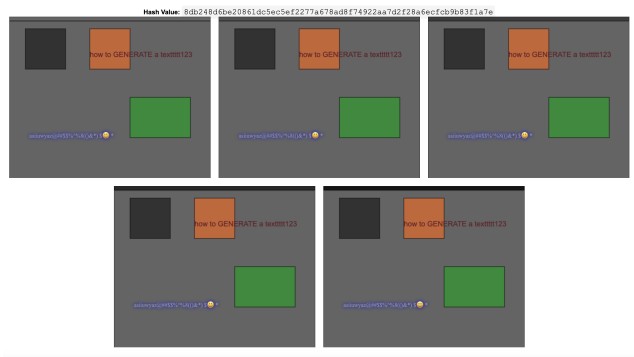

Figure 3: A page illustrates the attack on the Farbling mechanism, featuring five Filled Canvases, each topped with a uniquely colored rectangle.

Figure 4: An example illustrates the perturbation of the Green channel for a specific pixel across five canvases. Three canvases retain a value of 179, while two show a perturbed value of 180. By employing a majority voting technique, we recover the original Green channel value of 179.