# OpenReview forum: "Breaking the Shield: Analyzing and Attacking Canvas Fingerprinting Defenses in the Wild"
_ACM.org/TheWebConf/2025/Conference — WWW 2025 Poster_

### Official Review · Reviewer_xMts · 2024-11-29

**Novelty:** 2
**Technical Quality:** 2

**Review:**

Summary:
The paper evaluates defenses against canvas fingerprinting, a pervasive online tracking method, and introduces attacks targeting randomization-based mechanisms in browser extensions. It demonstrates vulnerabilities in current defenses, providing recommendations to enhance privacy protections.

Strength:
1. Comprehensive analysis of defenses with 18 browser extensions and 5 major browsers, comparing defense techniques such as randomization, blocking, and filter lists.
2. Disclosed recommendation to the Brave browser.

Weakness:
1. The inclusion of Tor in the analysis of popular browsers lacks a clear rationale, especially since Tor’s primary focus is anonymity rather than mainstream usage, which may skew comparisons against browsers like Chrome or Firefox.
2. The paper discusses canvas randomization as a defense mechanism but does not provide concrete examples of its application in real-world scenarios or how randomization affects usability and security outcomes, leaving its practical implications unclear.
3. While the paper acknowledges usability issues caused by canvas randomization, such as visual distortions or functionality loss, it does not provide quantitative analysis to balance these trade-offs against security improvements, making it difficult to assess the practicality of implementing randomization defenses.
4. Section 3 appears to present background information about existing defense strategies, but its placement and tone may cause confusion, as readers might mistakenly interpret it as the authors’ proposed design; moving this to the background section would improve clarity.
5. The paper does not effectively articulate its unique contributions or the motivation for revisiting canvas fingerprinting defenses, which weakens its novelty and makes the work appear incremental rather than groundbreaking.
6. The study compares various browser extensions but does not evaluate their performance relative to the browsers’ default protections, such as Chrome’s built-in anti-fingerprinting mechanisms, which limits the ability to assess whether extensions provide significant privacy enhancements.
7. The transitions between sections are unclear, making the overall narrative difficult to follow; for example, the discussion of current defenses in Section 4 does not smoothly connect to the results in Section 5, and Sections 6 and 7 introduce technical details without adequate framing, which reduces readability and coherence.
8. The examples in Figures 2–4 display numerical results of successful attacks but fail to provide detailed insights into how these attacks work or their implications, focusing instead on pixel-level differences without interpretive context, which limits the practical significance of the findings.
9. The paper relies heavily on previously documented vulnerabilities in randomization techniques, offering incremental findings rather than novel approaches, and its recommendations, such as applying established methods to Brave, do not significantly advance the field.
10. While the paper exposes flaws in specific defenses, it does not explore their broader impact on large-scale tracking ecosystems or systemic privacy risks, missing an opportunity to contribute to a deeper understanding of tracking at scale.
11. The study focuses narrowly on randomization-based techniques and does not explore alternative methods, such as machine learning-based defenses or hybrid approaches; furthermore, the experiments are conducted in controlled, synthetic scenarios without validation in large-scale or real-world environments, which limits the generalizability and applicability of the findings.


Minor comments:
1. Different fonts are used in quotes and double quotes. Please check them for consistency.

**Questions:**

1. How effective are the current defense mechanisms in preventing fingerprinting, and to what extent do they compromise usability? A detailed evaluation of both security improvements and usability trade-offs would provide a clearer picture of their practicality.
2. Why were these five browsers chosen for the analysis? If they represent the most widely used browsers, providing usage statistics would strengthen the rationale. Alternatively, if the selection is based on criteria like best practices or specific features, please clarify the reasoning behind this choice.

**Reviewer Confidence:**

3: The reviewer is confident but not certain that the evaluation is correct

**Scope:**

3: The work is somewhat relevant to the Web and to the track, and is of narrow interest to a sub-community

---

### Official Review · Reviewer_qrUq · 2024-11-30

**Novelty:** 3
**Technical Quality:** 3

**Review:**

This paper investigates canvas fingerprinting defense technologies, analyzing 18 browser extensions and built-in protection measures from 5 mainstream browsers (Chrome, Firefox, Brave, Tor, and Safari). The study found that randomization techniques are the most widely adopted defense mechanism. The authors successfully launched attacks (pixel recovery and statistical attacks) against randomization-based defense mechanisms, including nine extensions and Brave browser's "Farbling" mechanism, revealing existing vulnerabilities. The paper concludes by proposing recommendations to strengthen existing defenses, including ensuring the unpredictability and irreversibility of randomization and implementing API call monitoring mechanisms.

### Strengths
1. **Comprehensive Defense Technology Analysis**: Conducted a broad and in-depth survey of existing canvas fingerprinting defense technologies, covering multiple standard defense methods such as blocking API access, modifying canvas content, using filter lists and machine learning models, providing readers with a comprehensive view of defense technologies in this field.

2. **Effectiveness Evaluation in Practical Applications**: We evaluated these defense technologies' effectiveness through experiments on real browser extensions and mainstream browsers, making the research results more realistic and credible. The experimental results demonstrated vulnerabilities in some extensions and browser defense measures, providing strong evidence for subsequent improvements.

3. **Introduction of New Attack Methods**: Proposed two new attack methods - pixel recovery and statistical attacks - targeting widely used output randomization techniques. We have successfully applied these methods to multiple extensions and Brave browser's "Farbling" mechanism, revealing the fragility of these defense measures and providing new research directions for security researchers.

## Weakness Analysis
### (I) Technical Contributions
1. Although two new attack methods were proposed, these attacks primarily target specific implementation vulnerabilities in existing defense technologies, with limited innovation in fundamental theory or technical principles. For instance, pixel recovery attacks rely on discovering specific patterns in the randomization process, while statistical attacks utilize majority voting techniques to recover original content. These methods may become ineffective when facing slight changes in defense mechanisms or more complex randomization strategies.

2. The paper primarily focuses on canvas fingerprinting defense technologies, with insufficient consideration of collaborative defense or interaction effects with other browser fingerprinting technologies (such as user agent strings, font fingerprinting, etc.). In practical applications, attackers often employ multiple fingerprinting techniques comprehensively. Improvements in canvas fingerprinting defense alone may fail to counter comprehensive fingerprinting attacks effectively.

### (2) Generality and Extensibility
1. The browser extensions and versions analyzed in the research are limited and may not represent all possible configurations and usage scenarios. With the release of new browser versions and extension updates, defense mechanisms may change, potentially affecting the applicability of the attack methods and conclusions. For example, the paper did not explore the effectiveness of these defense technologies and attack methods in mobile or niche browsers, limiting the generalizability of the research results.

2. The paper did not investigate the impact of emerging web technologies and application scenarios (such as WebAssembly, Progressive Web Apps, etc.) on canvas fingerprinting defenses or how to extend existing defense technologies to adapt to these new technologies. These new technologies might introduce attack surfaces or alter the effectiveness of existing defense technologies, potentially rendering the paper's conclusions insufficient when facing future technological developments.

### (3) Detailed Review Comments
1. **Limitations of Experimental Environment and Samples**: Experiments were primarily conducted on limited test pages and specific environments, potentially unable to cover all possible canvas usage scenarios and website interaction contexts. For instance, the effectiveness of defense technologies and attack methods was not fully validated for web applications highly dependent on canvas with complex interaction logic (such as online graphic editing tools and real-time games). Moreover, the number and diversity of samples used in the experiments might be insufficient to represent the actual canvas fingerprinting situation in the entire web ecosystem, leading to certain biases in research results.

2. **Insufficient Analysis of Privacy and Usability Trade-offs**: Although the paper mentioned trade-offs between different defense technologies regarding privacy and usability, it lacks in-depth analysis and quantitative assessment of how to ensure adequate defense while minimizing impact on user experience. For example, when discussing defense technologies like blocking API access and fixed output and their potential destruction of website functionality, the paper did not provide specific data or cases to illustrate the extent of such impacts or how to optimize them in practice.

3. **Insufficient Discussion of Dynamic Adaptability in Security Defense**: The web environment is dynamically changing, with attackers continuously improving attack methods and new fingerprinting technologies emerging. While the paper's defense recommendations have some guiding significance, they lack discussion on how defense technologies can dynamically adapt to new threats. For instance, how to automatically update randomization algorithms, timely adjust filter lists, or improve machine learning models to address new attack types was not adequately elaborated.

4. **Insufficient Emphasis on User Education and Awareness**: User education and awareness are equally important in addressing privacy threats like canvas fingerprinting. The paper only focused on technical defense and attacks without mentioning how to raise user awareness about canvas fingerprinting risks or guide users in correctly selecting and configuring privacy protection tools. Users might easily bypass or turn off effective defense measures without understanding, thus reducing overall privacy protection levels.

### (4) Innovation
As previously mentioned, the new attack methods primarily target implementation vulnerabilities in existing technologies. The innovation is relatively limited in the broader privacy protection technology field, failing to propose new theoretical frameworks or technical paradigms to address canvas fingerprinting and related privacy issues.

### (5) Rigor
1. some assumptions and calculation processes lack detailed explanation and verification in describing attack methods. For example, in statistical attacks, the assumption about the reliability of majority voting techniques is based solely on probabilistic calculations without considering noise, interference factors, and potential adversarial strategies in actual network environments. This might lead to differences between attack success rates and effectiveness in practical applications versus theoretical analysis.

2. In experimental evaluations, the paper merely pointed out their problematic nature for ineffective extensions or those failing to achieve expected protection effects without deeply analyzing the root causes of failure. This makes it difficult for readers to comprehensively understand the challenges these defense technologies might encounter during implementation and is not conducive to targeted improvements by subsequent researchers.

### (6) Relevance
The research primarily focuses on canvas fingerprinting defense in browser environments, with insufficient exploration of relevance to other privacy-related domains (such as mobile applications and IoT devices). In today's digital era, privacy protection needs span multiple platforms and devices. The research findings' application and promotion value in a broader privacy protection ecosystem remains to be further explored.

### (7) Verifiability
The paper provides specific details in experimental setup and attack method descriptions that are helpful for other researchers to reproduce. However, only the attack method code is planned for release at the code implementation level, without mentioning whether code for extension analysis and browser built-in protection mechanism testing will be publicly available. This might pose difficulties for other researchers in fully reproducing the experiments.

**Questions:**

1. Facing increasingly complex and diverse network attack methods and incredibly advanced persistent threats (APT) combining artificial intelligence and machine learning technologies, how can the defense technologies and recommendations in the paper effectively respond? Have you considered that these attacks might automatically adapt to and bypass existing randomization and monitoring mechanisms? If so, what specific countermeasure strategies and plans exist?

2. Considering significant differences in privacy and usability needs across user groups, how can a more personalized and adaptive canvas fingerprinting defense scheme be designed? For instance, for privacy-sensitive users (such as journalists and activists) and average internet users, how can different levels of usability be guaranteed while maintaining privacy protection? Are there related technical approaches or frameworks to implement such personalized defense?

**Reviewer Confidence:**

4: The reviewer is certain that the evaluation is correct and very familiar with the relevant literature

**Scope:**

3: The work is somewhat relevant to the Web and to the track, and is of narrow interest to a sub-community

---

### Official Review · Reviewer_Sjrk · 2024-12-02

**Novelty:** 5
**Technical Quality:** 5

**Review:**

This paper studies the robutness of current defenses against canvas fingerprinting. Four main defense techniques have been considered, including (i) blocking canvas APIs altogether, (ii) altering the canvas output with randomization-based technique or by fixing the output, (iii) leveraging filter lists of known fingerprinting libraries, (iv) or machine learning to detect potentially unknown and obfuscated codes.
Specifically, the authors successfully demonstrate bypasses against the randomization-based technique employed by 9 (7 on Chrome and 3 on Firefox) extensions and the custom "Farbling" implementation of Brave browser. The authors observed that, the randomization-based implemention of extensions typically consist in applying uniformely a unique-session-based perturbation vector to varying sections of canvas created in webpages. By employing a dummy base canvas with uniform pixel values, the authors  reverse-enginnered the initial purturbation vector, which can then be applied to an effective tracking canvas to recover its original output. Unlike the uniformity of the extensions perturbation vector, the Brave custom farbling method further adds randomness, which the authors nonetheless defeated with a "Statistical attack" consiting in employing 5 effective tracking canvas, and mojority voting to reconstruct unperturbed canvas outputs.



## Pros
- The paper evaluates the implementation of different canvas fingerprinting methods provided by browsers vendors, or web extensions.
- The pixel-recovery attack was mounted to defeat randomization-based techniques by browser extensions
- The sophisticated farbling technique of Brave browser can be bypassed with statistical attacks


## Cons
- The empirical study, in particular on browser extensions, is not systematic.
- The paper over-claims the results in this work based on unbacked assumptions, in particular regarding ML methods, and to a lesser extent, the filter-based methods
- The claim that there is no currently effective solution against canvas fingerprinting is too strong and not backed-up. Indeed, the authors mention that the ML algorithms are effective, at even finding previously unknown libraries. The sole criticims against them seems to be their performance, but the authors do not give any intuition on how poor the ML techniques perform.

## Minor
Typo in Section 2, Paragraph 3: "...actively pIn additionrevent canvas fingerprinting..."
Typo in Section 4.3, Paragraph 4: "...the privacy-focused broswer Brave...."

**Questions:**

- How did you gather the list of 18 extensions? Why only 18, provided that Firefox and Chrome have thousands of extensions in their respective stores? Why did you exclude browsers like Opera or Microsoft Edge which also have extensions?
- Why did you consider the 18 extensions if only 13 of them worked as expected? One could claim that 100K extensions were considered in the first place, but only 13 worked as expected
- What are the overhead of the different attacks, in particular the statistical attack against Brave?
- What is the setup on the empirical evaluation, i.e., details of the sites with fingerprinting libraries visited; how the different results were confirmed?
- How does the ML-based solutions compare to others in terms of performance overhead?

**Reviewer Confidence:**

4: The reviewer is certain that the evaluation is correct and very familiar with the relevant literature

**Scope:**

4: The work is relevant to the Web and to the track, and is of broad interest to the community

---

### Official Review · Reviewer_Xu89 · 2024-12-02

**Novelty:** 3
**Technical Quality:** 4

**Review:**

Summary:

This paper explores the effectiveness and vulnerabilities of canvas fingerprinting defenses across popular browsers and extensions. Canvas fingerprinting is a technique that uses subtle rendering differences in the `<canvas>` element to track users without their consent. In response, multiple defenses, including randomization and blocking of the Canvas API, have been proposed. The study examines 18 browser extensions and built-in defenses from major browsers (Chrome, Firefox, Brave, Tor, and Safari).

The paper highlights that randomization-based defenses are the most widely adopted, particularly in Brave's "Farbling" mechanism. However, the authors demonstrate that these defenses are not entirely effective, as they are susceptible to attacks such as the Pixel-Recovery Attack and Statistical Attack. These attacks exploit predictabilities in the randomization process, enabling the recovery of original canvas data. The study reveals that none of the evaluated defenses, including Brave's Farbling, provide an entirely secure solution to canvas fingerprinting.

Pros:
- Timely Topic
- Identification of vulnerable defenses

Cons:
- Limited novelty of proposed attacks
- Lack of systematic approach

Detailed Comments:

Thanks for the submission. I appreciate the authors’ efforts in evaluating existing canvas fingerprinting defenses. Fingerprinting attacks and defenses have garnered increasing attention in recent years due to growing concerns over user privacy. However, I have some concerns regarding the paper, as detailed below.

1. Limited novelty of the proposed attacks: While I acknowledge the importance of highlighting the implementation issues in existing defense extensions and browsers, the two attacks proposed in the paper seem quite specific to vulnerable implementations. From a defender’s perspective, these attacks also have relatively simple countermeasures. For example, the Pixel-Recovery Attack could be easily mitigated by modifying the scope of the RGBA tuple from being per-session to per-API call. The Statistical Attack appears more like a vulnerability report, as it targets specific weaknesses in Brave's implementation. It also seems like a vulnerability that could be addressed by adding a random or secret string to the SHA256 function arguments. In other words, the attacks presented in this paper are somewhat ad hoc and not necessarily offering new attack surfaces or systematically exploring existing attack surfaces and their associated costs. Given that the paper reveals that some widely-deployed defenses are ineffective, I would recommend focusing more on systematically analyzing the vulnerabilities within these defenses, rather than proposing new attacks that can be easily defended with relatively simple changes in code. Achieving perfect defense would require the canvas API to produce the same output across different devices, which is currently impossible without significant performance overhead. Could you clarify the key difficulties in achieving perfect defenses? It would be helpful to identify these challenges, assess how difficult they are to overcome, and discuss the trade-offs between security and usability in different scenarios. This would provide clearer insights into where improvements are feasible and what compromises are acceptable.

2. Unclear data collection process: In Section 4.1, the data collection process should be more clearly explained. For instance, the phrase "such as" should be followed by a list of all the keywords used, which could be included in the appendix. Additionally, specific criteria should be provided for determining whether an extension is designed for canvas fingerprinting defense. I performed a search for "canvas fingerprinting protections" on the Chrome Store and found an extension called “All Fingerprint Defender,” which is not listed in Table 1. While the data presented in the paper are not necessarily required for reproducibility, the steps involved in the data collection process should be clear and easily reproducible by others.

I would suggest that the authors delve deeper into analyzing the challenges in defending against canvas fingerprinting, particularly from an API or hardware-level perspective. By gaining insights into the underlying reasons for these difficulties, the authors could potentially identify a more balanced compromise between user experience and security. A more systematic proposal of a new, effective defense solution, based on these insights, would make a significant contribution. Additionally, evaluating this new solution with the attacks proposed in the paper would provide a more comprehensive analysis and strengthen the paper's contribution. This approach could help the research gain wider recognition and appreciation within the community.

**Questions:**

1.Could you clarify the fundamental difficulties in designing defenses for canvas fingerprinting?
2. Have you explored more generalized vulnerabilities in canvas fingerprinting defenses that might offer new attack surfaces?

**Reviewer Confidence:**

4: The reviewer is certain that the evaluation is correct and very familiar with the relevant literature

**Scope:**

4: The work is relevant to the Web and to the track, and is of broad interest to the community

---

### Official Review · Reviewer_WfMv · 2024-12-02

**Novelty:** 4
**Technical Quality:** 3

**Review:**

This paper investigates the effectiveness of various canvas fingerprinting defense mechanisms implemented in browser extensions and major browsers. The study critically evaluates defense techniques such as randomized output, fixed output, API blocking, and filter lists, while proposing two novel attack vectors: Pixel-Recovery Attack and Statistical Attack. These attacks successfully exploit vulnerabilities in popular randomization-based defenses, revealing significant gaps in their robustness. The authors conclude by offering recommendations for improving canvas fingerprinting defenses and disclosing their findings to affected stakeholders.

While the paper briefly mentions usability trade-offs, it does not provide an in-depth analysis of the balance between privacy and functionality for end-users. The potential of machine learning as a defense mechanism is mentioned but not explored in depth, leaving a gap in the evaluation of adaptive defenses. The broader implications of the proposed attacks on other tracking techniques beyond canvas fingerprinting are not fully addressed.

**Questions:**

1. Can you provide more detailed insights into the usability trade-offs associated with enhanced privacy measures like strict API blocking or aggressive randomization? For instance, how do these measures impact performance, user experience, or compatibility with modern web applications?

2. While machine learning is mentioned as a potential adaptive defense, could you elaborate on its feasibility and effectiveness in detecting and mitigating advanced canvas fingerprinting techniques?

3. Do the proposed attack methodologies, such as Pixel-Recovery and Statistical Attacks, have broader applicability to other fingerprinting techniques, like WebGL or font-based fingerprinting? If so, what additional mitigations might be required to address these vulnerabilities comprehensively?

**Reviewer Confidence:**

3: The reviewer is confident but not certain that the evaluation is correct

**Scope:**

4: The work is relevant to the Web and to the track, and is of broad interest to the community